# Predicting Risk of Ammonia Exposure in Broiler Housing: Correlation with Incidence of Health Issues

**DOI:** 10.3390/ani14040615

**Published:** 2024-02-14

**Authors:** Leonardo V. S. Barbosa, Nilsa Duarte da Silva Lima, Juliana de Souza Granja Barros, Daniella Jorge de Moura, Fernando Estellés, Adrian Ramón-Moragues, Salvador Calvet-Sanz, Arantxa Villagrá García

**Affiliations:** 1College of Agricultural Engineering, State University of Campinas, 501 Candido Rondon Avenue, São Paulo 13083-875, Brazil; valentino.leo@icloud.com (L.V.S.B.); jdsgbarros@gmail.com (J.d.S.G.B.); 2Department of Animal Science, Federal University of Roraima, Boa Vista 69300-000, Brazil; nilsa.silva.lima@gmail.com; 3Institute of Animal Science and Technology, Universitat Politècnica de València, Camino de Vera s.n., 46022 Valencia, Spain; feresbar@upv.es (F.E.); armzulu@gmail.com (A.R.-M.); salcalsa@upvnet.upv.es (S.C.-S.); 4Centro de Investigación en Tecnología Animal (CITA), Valencian Institute for Agricultura Research (IVIA), 12400 Segorbe, Spain; villagra_ara@gva.es

**Keywords:** ammonia, machine learning, chicken production

## Abstract

**Simple Summary:**

This study assesses the risk of ammonia exposure in broiler chicken production and correlates these risks with health issues, utilizing machine learning techniques. Two broiler breeds, fast-growing (Ross^®^, 42 days) and slow growing (Hubbard^®^, 63 days), were studied at different densities. Slow-growing birds had a fixed density of 32 kg/m^2^, while fast-growing ones were housed at low (16 kg/m^2^) and high (32 kg/m^2^) densities. The high concentration of atmospheric ammonia has been associated with a greater occurrence of bird health problems, such as pododermatitis, visual impairment and mucosal lesions compared to birds stocked in controlled environments with low concentrations of atmospheric ammonia. A total of 1250 birds were used, and classification algorithms (decision tree, SMO, Naive Bayes, and Multilayer Perceptron) were applied to predict ammonia risk levels. The analysis involved data selection, pre-processing, transformation, mining, and interpretation of results. The Multilayer Perceptron proved the most effective in predicting exposure risk. The Spearman’s correlation coefficient indicated a strong correlation between high ammonia concentrations and higher incidences of injuries in the birds that were evaluated. This research highlights the importance of managing ammonia levels in broiler production to mitigate health risks for both fast- and slow-growing breeds.

**Abstract:**

The study aimed to forecast ammonia exposure risk in broiler chicken production, correlating it with health injuries using machine learning. Two chicken breeds, fast-growing (Ross^®^) and slow-growing (Hubbard^®^), were compared at different densities. Slow-growing birds had a constant density of 32 kg m^−2^, while fast-growing birds had low (16 kg m^−2^) and high (32 kg m^−2^) densities. Initial feeding was uniform, but nutritional demands led to varied diets later. Environmental data underwent selection, pre-processing, transformation, mining, analysis, and interpretation. Classification algorithms (decision tree, SMO, Naive Bayes, and Multilayer Perceptron) were employed for predicting ammonia risk (10–14 pmm, Moderate risk). Cross-validation was used for model parameterization. The Spearman correlation coefficient assessed the link between predicted ammonia risk and health injuries, such as pododermatitis, vision/affected, and mucosal injuries. These injuries encompassed trachea, bronchi, lungs, eyes, paws, and other issues. The Multilayer Perceptron model emerged as the best predictor, exceeding 98% accuracy in forecasting injuries caused by ammonia. The correlation coefficient demonstrated a strong association between elevated ammonia risks and chicken injuries. Birds exposed to higher ammonia concentrations exhibited a more robust correlation. In conclusion, the study effectively used machine learning to predict ammonia exposure risk and correlated it with health injuries in broiler chickens. The Multilayer Perceptron model demonstrated superior accuracy in forecasting injuries related to ammonia (10–14 pmm, Moderate risk). The findings underscored the significant association between increased ammonia exposure risks and the incidence of health injuries in broiler chicken production, shedding light on the importance of managing ammonia levels for bird welfare.

## 1. Introduction

Atmospheric emissions, particularly ammonia (NH3) emissions from agricultural activities, pose significant challenges to agricultural systems [1]. Monitoring and mitigating ammonia emissions demand precise assessments of ventilation systems, continuous detection during production, and effective strategies [2]. Notably, variations in emissions exist across houses, system types, bird ages, and breeds [2,3].

Poultry facilities predominantly emit ammonia due to the microbial degradation of uric acid in poultry manure, making it a primary pollutant gas [2,4]. Key factors influencing NH_3_ emissions include temperature, moisture content, pH, ventilation rates, litter management, and composting methods. The resulting ammonia deposition can lead to environmental damage through direct toxicity, although effective mitigation strategies are available [1].

Determining ammonia emissions, especially in facilities with natural ventilation, is challenging. Accurate assessment of ventilation rates becomes crucial, impacting the prediction of emissions concerning gas concentration and ventilation rate [5].

For poultry production, precise quantification, and analysis of ammonia concentrations during the production process are paramount. Additionally, obtaining accurate estimates of emissions and concentrations across different poultry breeds is essential to evaluate their impact on bird performance and health.

High stocking density means more birds per area, which can impact animal welfare and living conditions. In Europe, standards vary, but in some cases, lower densities are sought to promote the comfort and health of the birds. The exposure of animals to high levels of ammonia can lead to irritation of mucous membranes and the respiratory tract, conjunctivitis, and dermatitis [6,7,8,9].

The experiment was conducted in accordance with EU animal research regulations, with protocol number 2018/VSC/PEA/0067. The test was carried out at the Animal Technology and Research Center (CITA-IVIA), located in Segorbe (Castellón, Spain).

The study aimed to forecast ammonia exposure risk in broiler chicken production, correlating it with health injuries using machine learning.

## 2. Materials and Methods

### 2.1. Experiment

An animal experiment was carried out during the winter in Spain in 2019 to predict the risk condition of ammonia levels and correlate them with the health risk of broiler chickens. The experiment was approved by the Animal Ethics Committee N2018/VSC/PEA/0067 on 16 April 2018. This study was part of a doctoral thesis.

Two identical rooms (Room 1 and Room 2) were used in this test, measuring 13.2 m × 5.95 m, totaling approximately 70 m^2^ for each room. An automated temperature control system was installed (DNP Climate Controller, Exafan, Spain), which controlled ventilation rates in accordance with commercial temperature recommendations. The room temperature was gradually decreased from 32 °C (day 1) to 19 °C (day 42). Temperature was controlled using the temperature control sensor and recorded along with relative humidity every 10 min using a data logger (HOBO U12, Onsetcomp, Bourne, MA, USA). Furthermore, each room was equipped with an electrochemical NH_3_ sensor (DOL 53, Dräger, Germany). Room 1 was programmed to maintain a maximum of 10 ppm of NH_3_, while Room 2 was programmed to maintain a maximum of 20 ppm. These environmental conditions (ammonia concentration) were programmed to be maintained from the fourth week onwards, that is, in the second half of the production cycle, when ammonia levels tend to be higher within broiler production systems. A propane heater was used to maintain an adequate room temperature.

The difference in ammonia concentration between rooms was obtained through ventilation. However, the different ventilation rate was not significant enough to cause large temperature differences between the rooms. For the initial four weeks, both rooms maintained a comparable temperature. Yet, starting from week 5, Room 1 elevated ventilation rates led to an average temperature reduction of 1.7 °C. It is worth noting that this 1.7 °C difference is an overall average across all weeks studied. In the crucial first two weeks, the difference did not surpass 0.3 °C, making the temperature conditions practically identical and unlikely to impact the birds’ performance. Relative humidity remained consistent between the two rooms.

The experiment was carried out during the winter period, when gas concentrations were expected to be higher due to lower ventilation rates. To ensure that the desired concentrations were achieved during a relevant part of the poultry production period, it was decided to apply a urea solution to the litter. The dosage was always 0.21 L/m^2^ of urea solution, with a concentration of 187.5 g/L on day 32 of the rearing cycle and 93.75 g/L on days 39, 51, and 56.

Two commercial lines of broiler chicken were used, one with fast-growing (Ross^®^, slaughter age 42 days) and another with slow-growing (Hubbard^®^, slaughter age 63 days). All slow-growing birds were housed at a density of 32 kg m^−2^. Fast-growing birds were housed in two different housing densities: low stocking density with a final stocking density of 16 kg/m ^2^ and high density with a final stocking density of 32 kg m^−2^. A total of 1250 birds were used in this experiment, 450 of which were fast-growing birds and 800 of which were slow-growing birds. A total of 102 birds were allocated into 6 boxes designated for the slow-growth treatment at high stock density, resulting in 17 birds per box. Similarly, another 102 birds were placed in 6 boxes for the fast-growth treatment at high stock density, also with 17 birds per box. Lastly, an additional 102 birds were distributed across 6 boxes for the fast-growth treatment, housed at high stock density, maintaining the ratio of 17 birds per box. The remaining birds were positioned in the external corridors of the boxes to simulate real-field breeding conditions. Each box had 3 nipple drinkers and a manual feeder. The dimensions of the high-density boxes were 1 × 1.3 m^2^, whereas the low-density boxes, stocking the same number of animals, were more spacious, with dimensions of 2 × 1.3 m^2^ (Figure 1).

All animals were fed with Nanta brand starter commercial feed for broiler chickens from day 1 to day 18 of the trial. From day 18 until the end of the trial, each strain was fed with different feeds. The feed transition was done gradually over two days, mixing the two feeds to enhance adaptation. For feeding fast-growing breed (treatments HD and LD), two commercial feeds for fast growth (Nanta A80) and slow growth (Nanta A32) were used.

Previously, centesimal chemical analysis was conducted. The composition obtained for the fast-growing breed was crude protein: 18%, crude fat: 1.9%, crude fiber: 3.1%, ash: 6.2%, calcium: 1.00%, phosphorus: 0.67%, sodium: 0.16%, methionine: 0.33%, lysine: 0.94%. for the slow-growing breed: crude protein: 15.7%, crude fat: 3.9%, crude fiber: 2.6%, ash: 5.8%, calcium: 1.00%, phosphorus: 0.67%, sodium: 0.16%, methionine: 0.28%, lysine: 0.82%.

All birds randomly stocked in the boxes were weighed (kilograms) weekly, and their respective feed consumption was calculated. Average daily weight gain (ADG) per bird was calculated for each week of rearing, and average daily weight gain accumulated over the entire study period (42 days for fast-growing birds and 63 for slow-growing birds). Feed conversion (FCR) was also obtained for each week and for accumulated period, dividing the amount of food consumed by each pen by the weight gain of all birds present in it Table 1. The data in Table 1 are from a previously published study [10].

Room ammonia concentration data was collected by installing electrochemical sensors (DOL 53, Dräger, Germany) in each room. Assessments were carried out 24 h a day, every day of the week. In addition to the productive character determinations, during the development of the experiment, animals were sacrificed, and samples were taken at four moments: day 0, day 21, day 42, and day 63 (day 63 only for animals from the slow-growing lineage). All the sacrificed birds were previously stunned by an electric shock.

In the first sampling, on day 0 of the study, 30 animals from each lineage were randomly sacrificed before distribution into the boxes. In the second and third sampling, days 21 and 42 of the experiment, respectively, 5 animals were sampled from each pen, resulting in a total of 180 animals for each sampling day. After sacrifice, the animals were necropsied.

The indicators were assessed following the necropsy protocol, conducted by the veterinarians in our research team. Necropsy procedures were consistently deliberated upon by the same researchers throughout the study.

The fourth and final sampling was carried out on day 63 of the experiment with the same procedure as the previous two but involving only animals from the slow-growing lineage, since animals from the fast-growing lineage have a commercial production cycle of 42 days.

In the last two collections, at 42 days for fast-growing birds and at 63 days for fast-growing birds, the sampled animals were inspected for symptoms related to prolonged exposure to NH_3_. These exams aimed to find epidermal lesions on the legs and injuries to the eyes and respiratory tract due to this irritating gas.

### 2.2. Data Mining Approach

The following data analysis steps were performed: data selection, pre-processing, transformation, mining, analysis, and interpretation of results.

The data preprocessing stage covers data understanding and data preparation, which includes standardizing nomenclatures, cleaning the raw data in the spreadsheet, and dividing the database in the Weka software (version 3.8.4) [11].

For training and testing, the model employed a “stratified remove folds” filter to bifurcate the dataset into training and testing subsets. Pre-processing also included the discretization of attributes into classes that reduces and simplifies the data, making learning faster and the results denser, according to the proposed methodologies [11,12].

In the processing stage, the data set was analyzed by applying predictive classification models for training (75% of the data set with 17,062 instances) and for validation of the model with the test set (25% of the data set with 5688 instances).

The classification algorithms, decision tree (J48), Sequential Minimal Optimization (SMO), Naive Bayes and Multilayer Perceptron were applied to the training and test data sets to build a rule model for predicting ammonia risk levels in broiler chickens. The cross-validation technique (test mode: 10-fold cross-validation) was used to parameterize the analysis in all models. The number of attributes used in the modeling was seven, including “housing_condition”, “age_week”, “T-hobo (temperature of hobo)”, “UR%”, “Vent (ventilation)”, “NH3_ppm” and the response attribute “Ammonia_concentration_risk”, with a total of 5688 instances.

The study developed a machine learning model to predict the risk condition of ammonia concentration in the production of chickens of slow and fast-growing breeds with low and high production densities. The study also compared the performance of all algorithms with respect to their prediction abilities and model quality. When evaluating the models, the data was divided into training and testing subsets, then the results were compared by the performance metrics of the algorithms. The flowchart used to identify the best test algorithm is shown in Figure 2 (adapted from [13]).

The criterion used to discretize the classes of the response attribute (target) “ammonia concentration risk condition” included ammonia concentration levels in five classes described in Table 2 [9,14,15]. 

### 2.3. Performance Measures of Classification Models

The performance of the models was evaluated using various metrics, including accuracy, incorrectly classified instances, Kappa statistics, true positive rate, false positive rate, precision, sensitivity (recall), F value, Matthews Correlation Coefficient (MCC) and the confusion matrix [16,17].

Below are the equations used to evaluate the performance of the algorithms for accuracy, precision, sensitivity (recall), Matthews correlation coefficient (MCC) and F value calculated from Equations (1) to (6), respectively:(1)$$False\;Positive\;Rate=1-\frac{\left(TN\right)}{\left(FP+FN\right)}$$
(2)$$Accuracy=\frac{\left(TP+TN\right)}{\left(TP+TN+FP+FN\right)}$$
(3)$$Precision=\frac{\left(TP\right)}{\left(TP+FP\right)}$$
(4)$$Sensitivity=\frac{\left(TP\right)}{\left(TP+FN\right)}$$
(5)$$MCC=\frac{TP\times TN-FP\times FN}{\surd (TP+FP)(TP+FN)(TN+FP)(TN+FN)}$$
(6)$$F\;value=2\times \left\lfloor \frac{\left(Precision\times Sensitivity\right)}{\left(Precision+Sensitivity\right)}\right\rfloor$$
where TP: true positive; TN: true negative; FP: false positive; FN: false negative.

The following items are calculated in the confusion matrix. true positives (TP), which are the positive tuples that were correctly labeled by the classifier; true negatives (TN), which are the negative tuples that were correctly labeled by the classifier; false positives (FP), which are negative tuples that were incorrectly labeled as positive; and false negatives (FN), which are the positive tuples that have been mistakenly labeled as negative. This shows the relationship between observed and predicted values in a classification problem [17].

The study compares the performance of all algorithms with respect to their prediction abilities and model quality. The flowchart used to identify the best training algorithm is shown in Figure 2.

### 2.4. Spearman Correlation Analysis

From the ammonia concentration risk classification data (target attribute “Ammonia_concentration_risk”), only those presenting some degree of risk were considered. The Spearman correlation coefficient (ρ, rho) was calculated, considering the presence values (numerical counts) of diseases and injuries quantified during the experimental phase, including pododermatitis, vision/affected, and mucosal injury, which encompass assessments of trachea, bronchi, lungs, eyes, paw injury, and other injuries.

A non-parametric correlation measure was applied to the injury incidence data as a function of the ammonia risk level (1 and 10 ppm) with the aim of correlating the injury incidence and the ammonia level in the conditions studied, calculated from Equation (7).
(7)$$r_{s}=1-\frac{6\sum d_{i}^{2}}{n(n^{2}-1)}$$

## 3. Results

The overall performance of the models indicated an accuracy of 100% for J48, 91.58% for SMO, 92.44% for Naive Bayes, and 99.05% for Multilayer Perceptron. The biggest error in classification was for the SMO model. The Kappa statistic was also 100% for the J48 model, followed by 89.12% for the SMO model, 90.28% for the Naive Bayes model, and 98.77% for Multilayer Perceptron. The overall performance results indicate that the Multilayer Perceptron model was the best classification model for detecting risk from ammonia concentration in chicken production.

The visualization of the decision tree generated by the J48 classification model is shown in Figure 3, the scheme indicates that if the ammonia concentration is >9 ppm, the ammonia concentration must be observed; when this concentration is ≤ 14: o, the risk is moderate; when the ammonia concentration is >4 ppm, the ammonia concentration must be observed; if the ammonia is ≤20 ppm, the risk is high; and if it is >20 ppm, the risk is very high for birds.

The main results presented in Table 3 and Table 4 show that the J48 model generated good average performance (100%) in all metrics; the SMO model presented similar performance, but with a recall of 77% for the “low risk” class. The Naive Bayes model presented similar results, with values above 85% in all metrics, and the Multilayer Perceptron model obtained the best performance of all with the most adjusted metrics, as occurred with J48. However, these overfitted results may contain overfitting.

The models’ confusion matrix is shown in Table 5. The J48 model obtained 100% correct answers for all classes and did not present a classification error. The SMO and Naive Bayes models showed more classification errors than J48 and Multilayer Perceptron, indicating that these models can still be adjusted to increase accuracy per class. The smaller the error in classifying the risk of ammonia concentration in facilities, the better the possibility of managing this ammonia concentration when decision-making is required.

Spearman’s correlation assessed the interrelationship between the risk variable of exposure to ammonia (10–14 pmm, Moderate risk) and the incidence of diseases resulting mainly from ammonia gas (Table 6). Spearman’s correlation between the risk of exposure to ammonia and the incidence of lung health problems showed a correlation of 0.549, the risk for the bronchi 0.189, for the eyes 0.378, and for the paws 0.375, so if the value of ⍴ is approached 0, the association between the two intervals is weaker. The higher the absolute value of the coefficient, the stronger the relationship between the variables. The correlation involving other injuries caused by ammonia showed a strong correlation when compared to other types of injuries, indicating a greater association between the appearance of injuries when birds are exposed to higher levels of ammonia in the production process.

## 4. Discussion

When evaluating risk levels of ammonia concentration across various broiler production systems, including those with fast and slow-growing birds, a clear scenario emerges depicting the impact of ammonia concentration on the production process. A convergence is observed between ammonia concentration levels and the type of production systems, as well as the technology employed in both broiler and egg production. In the context of machine learning, Internet of Things (IoT) data has been successfully employed to predict injuries that negatively impact poultry production. Although there has been a steady increase in literature addressing applications of digital technology in agribusiness in recent years, there is a notable lack of peer-reviewed articles that focus on AI-enabled IoT systems in managing poultry health and welfare. Furthermore, most previous studies are limited to specific aspects of bird welfare [18].

Similarly to this study, researchers conducted monitoring and observations, covering variables such as temperature, humidity, feces content, ammonia levels, and humidity in the poultry environment [19,20].

Studies show the implementation of sensors to supervise and regulate environmental conditions, activating appropriate devices such as ventilation, lighting, refrigeration, and heating systems, as mentioned in previous research [21,22,23,24,25].

For instance, following the approach of this study to predict injury risks, a system was designed for automatically detecting sick broilers. This system, based on the ResNet residual network, achieved a remarkable 93.70% accuracy when monitoring the behavioral physiology and productive performance of meat birds. In this study, even greater accuracy was obtained in predicting injuries caused by ammonia, with values above 98% for the Multilayer Perceptron model [26].

A system with low computational complexity that demonstrated an accuracy of 80.00% was introduced. This system can automatically adjust the environmental behavior of birds, considering variables such as temperature, humidity, light intensity, and population density [27].

Initially, it is crucial to monitor environmental parameters on a poultry farm, including elements such as temperature, humidity, ammonia levels, and light. This monitoring is essential to ensure effective control of internal conditions by automation systems. Several Machine Learning techniques are employed to monitor these environmental parameters, ranging from linear regression to fuzzy logic neuro-fuzzy and neural networks, as well as deep learning [19,23,28].

A notable system was crafted in this study, attaining an accuracy of 97.00%. This system is designed to regulate hydrothermal parameters, encompassing temperature, relative humidity, and contaminating gases. This fosters the establishment of optimal conditions for effective poultry production. The developed model’s accuracy closely approached the values observed in this study, underscoring the efficacy of these models in predicting environmental conditions [23].

Moreover, a MultiBox Detector was implemented to automatically diagnose the health status of broiler chickens, achieving an outstanding average accuracy of 99.70%. This performance surpassed the accuracy of models examined in the ammonia risk prediction study [29].

In the realm of activity recognition, a comparative analysis among decision trees, Naïve Bayes, and neural networks was conducted to discern the activities of broiler chickens [30]. The results revealed that neural networks exhibited superior overall accuracy, reaching 82.10%. Similarly, the application of the classification tree algorithm to identify behaviors in broiler breeders yielded an overall success rate of 70.30% in the validation set. Mirroring these investigations, the research focused on predicting ammonia-related risks, where the Naïve Bayes model also demonstrated excellent results, surpassing 85% accuracy.

Given that ammonia is a primary air pollutant in poultry facilities, it has significant repercussions on the ecosystem, environment, bird welfare, and human health [15], accurately estimating NH_3_ concentration becomes an essential imperative. This precision is crucial for proper waste management, with the ultimate goal of preserving environmental health, human health, and animal welfare [15,24]. This is just one of numerous factors in poultry breeding. Identifying and, more importantly, addressing these factors is crucial to safeguard environmental health, human well-being, and animal welfare.

This research conducted a performance evaluation involving four models, namely: multilayer perceptron, adaptive neuro-fuzzy inference systems integrated with grid partitioning and subtractive clustering (ANFIS-GP and ANFIS-SC), as well as multiple linear regression analysis. The results highlighted that ANFIS-SC stood out as the most accurate, recording an R-squared value of 0.86 in the validation set. In this study, when performing the calculation for Spearman’s correlation, a greater correlation was observed between exposure to ammonia and the incidence of lung health problems, with a correlation of 0.549 [31].

In the context of estimating ammonia concentration in poultry farms, employed a subtractive clustering technique to determine the optimal input parameters in their regression model [31]. Furthermore, proposed a real-time segmentation algorithm based on K-means clustering and the ellipse model, aiming at automated diagnosis of the health status of broiler chickens [29].

The Multilayer Perceptron demonstrated remarkable performance, establishing itself as an excellent tool for predicting the risk of injuries in broiler chickens related to ammonia concentration.

## 5. Conclusions

Considering the accuracy by risk level of exposure to ammonia in the broiler chicken production process, including both fast and slow growth, the Multilayer Perceptron emerges as the best predictive model in terms of evaluation and performance.

A strong positive correlation was observed between the concentration of ammonia and the incidence of lung injuries (Spearman r = 0.55) and injuries in other areas of the birds’ bodies, such as air sacs and foot pads (Spearman r = 0.75).

The prediction model for the risk level of injuries based on ammonia concentration is important, as it complements the correlation results. This information is crucial for early decision-making regarding poultry management, waste management, and controlling the microclimate in the poultry farming environment.

## Figures and Tables

**Figure 1 animals-14-00615-f001:**
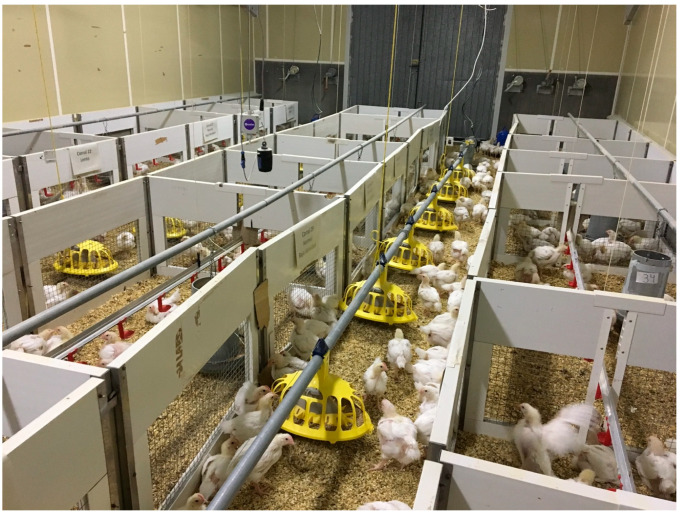
Experimentation room with detailed layout of experimental compartments.

**Figure 2 animals-14-00615-f002:**
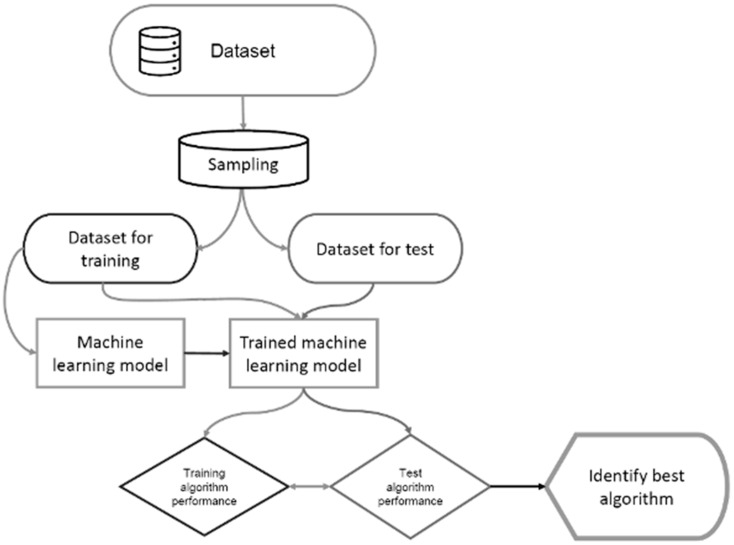
Flowchart of training and testing models to identify the best model.

**Figure 3 animals-14-00615-f003:**
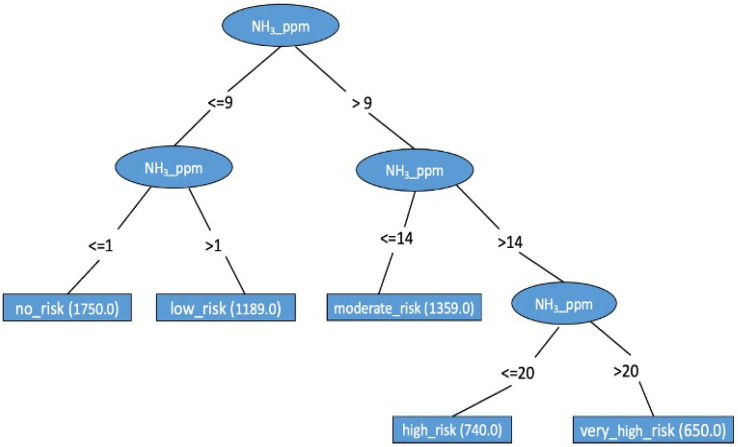
Visualization of J48 decision tree model.

**Table 1 animals-14-00615-t001:** Broiler growth and feeding data: Comparison between SHD, FHD and FLD treatments.

Day	SHD_Weight (g/bird)	FHD_Weight (g/bird)	FLD_Weight (g/bird)	SHD_Feed Consumption (g/animal/week)	FHD_Feed Consumption (g/animal/week)	FLD_Feed Consumption (g/animal/week)	SHD_Feed Conversion Ratio	FHD_Feed Conversion Ratio	FLD_Feed Conversion Ratio
7	144.8	157.75	160.45	0.11	0.134	0.13	1.14	1.2	1.12
14	323.1	413.2	428.6	0.26	0.371	0.36	1.58	1.35	1.38
21	544.55	788.25	789.15	0.5	0.713	0.665	2.08	2.02	1.89
28	907.3	1234.6	1233.55	0.67	1169.5	1086.5	1.88	2.66	2.46
35	1298.15	1810.05	1788.3	0.68	1448.5	1272.0	1.83	2.54	2.27
42	1785.05	2471.15	2461.1	0.9	1506.0	1362.5	1.89	2.25	2.09
49	2234.15			516.49			2.37		
56	2726.4			1198.5			2.45		
63	3201.45			1284.5			2.73		

SHD: slow-growing-high density. FHD: fast-growing-high density. FLD: fast-growing-low density. Adapted from: [10].

**Table 2 animals-14-00615-t002:** Criteria used for creating the target attribute risk of ammonia concentration for broiler.

Ammonia Concentration Risk	
Target Attributes	Risk Level
No risk	0–1 pm
Low risk	2–9 pm
Moderate risk	10–14 pm
High risk	15–20 pm
Very high risk	>21 pm

**Table 3 animals-14-00615-t003:** Overall performance of classification models J48, SMO, Naïve Bayes, and Multilayer Perceptron.

Classifier Models	J48 Tree	SMO	Naive Bayes	Multilayer Perceptron
Correctly classified instances (%)	100	91.58	92.44	99.05
Incorrectly classified instances (%)	00	8.42	7.56	0.95
Kappa statistic (%)	100	89.12	90.28	98.77

SMO: Sequential Minimal Optimization.

**Table 4 animals-14-00615-t004:** Performance of classification models J48, SMO, Naïve Bayes, and Multilayer Perceptron by ammonia concentration risk levels.

	**J48 Tree Model**					
**Accuracy by Class**	**TP Rate (%)**	**FP Rate (%)**	**Precision (%)**	**Recall (%)**	**F-Measure (%)**	**MCC (%)**
No risk	100	100	100	100	100	100
Low risk	100	100	100	100	100	100
Moderate risk	100	100	100	100	100	100
High risk	100	100	100	100	100	100
Very high risk	100	100	100	100	100	100
	**SMO Model**					
**Accuracy by Class**	**TP Rate (%)**	**FP Rate (%)**	**Precision (%)**	**Recall (%)**	**F-Measure (%)**	**MCC (%)**
No risk	97	1.0	98	97	97	96
Low risk	77	1.0	93	77	84	81
Moderate risk	94	6.0	83	94	89	85
High risk	95	2.0	87	95	91	90
Very high risk	92	0.01	99	92	95	95
	**Naive Bayes Model**					
**Accuracy by Class**	**TP Rate (%)**	**FP Rate (%)**	**Precision (%)**	**Recall (%)**	**F-Measure (%)**	**MCC (%)**
No risk	96	00	100	96	98	97
Low risk	88	2.0	93	88	90	88
Moderate risk	89	3.0	89	89	89	86
High risk	95	2.0	87	95	91	89
Very high risk	94	2.0	86	94	90	89
	**Multilayer Perceptron Model**					
**Accuracy by Class**	**TP Rate (%)**	**FP Rate (%)**	**Precision (%)**	**Recall (%)**	**F-Measure (%)**	**MCC (%)**
No risk	99	1.0	99	99	99	98
Low risk	98	1.0	98	98	98	98
Moderate risk	100	1.0	99	100	99	99
High risk	99	00	100	99	99	99
Very high risk	100	00	100	100	100	100

SMO: Sequential Minimal Optimization. TP Rate: true positive rate; FP Rate: false positive rate. MCC: Matthews correlation coefficient.

**Table 5 animals-14-00615-t005:** Confusion matrix of prediction models J48, SMO, Naïve Bayes, and Multilayer Perceptron.

J48 Tree Model						
No Risk	Low Risk	Moderate Risk	High Risk	Very High Risk	Total	Classified as
1750	0	0	0	0	1750	No risk
0	1189	0	0	0	1189	Low risk
0	0	1359	0	0	1359	Moderate risk
0	0	0	740	0	740	High risk
0	0	0	0	650	650	Very high risk
1750	1189	1359	740	650	5688	
**SMO Model**						
**No Risk**	**Low Risk**	**Moderate Risk**	**High Risk**	**Very High Risk**	**Total**	**Classified as**
1702	48	0	0	0	1750	No risk
41	918	230	0	0	1189	Low risk
0	20	1280	55	0	1355	Moderate risk
0	0	27	706	7	2095	High risk
0	0	0	51	599	650	Very high risk
1743	986	1537	812	606	5688	
**Naive Bayes model**						
**No Risk**	**Low Risk**	**Moderate Risk**	**High Risk**	**Very High Risk**	**Total**	**Classified as**
1685	59	0	0	6	1750	No risk
0	1046	135	0	8	1189	Low risk
0	20	1211	67	61	1359	Moderate risk
0	0	15	702	23	740	High risk
0	0	0	36	614	650	Very high risk
1685	1125	1361	805	712	5688	
**Multilayer Perceptron model**						
**No Risk**	**Low Risk**	**Moderate Risk**	**High Risk**	**Very High Risk**	**Total**	**Classified as**
1731	19	0	0	0	1750	No risk
22	1164	2	1	0	1189	Low risk
2	0	1357	0	0	1359	Moderate risk
0	0	8	732	0	2099	High risk
0	0	0	0	650	650	Very high risk
1755	1183	1359	733	650	5688	

**Table 6 animals-14-00615-t006:** Paired Spearman correlations.

Sample 1	Sample 2	Correlation	95% CI ρ	*p*-Value
Risk NH_3_ (ppm) 10–14 pmm	bronchi	0.189	−0.600; 0.792	0.654
Risk NH_3_ (ppm) 10–14 pmm	lungs	0.549	−0.313; 0.915	0.159
Risk NH_3_ (ppm) 10–14 pmm	eyes	0.378	−0.470; 0.863	0.356
Risk NH_3_ (ppm) 10–14 pmm	paws	0.375	−0.472; 0.862	0.360
Risk NH_3_ (ppm) 10–14 pmm	other injuries	0.750	−0.019; 0.961	0.032

Other lesions: air sacs, liver, and skin.

## Data Availability

The data presented in this study are available in the article.

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
