# Peer review of "Predicting Risk of Ammonia Exposure in Broiler Housing: Correlation with Incidence of Health Issues"

_animals, 2024, doi:10.3390/ani14040615_

Round 1
Reviewer 1 Report
Comments and Suggestions for Authors
Dear authors,
you have dealt with a very important and interesting topic. To make the procedure and the results more accessible to the reader, I have a few suggestions.
Introduction:
You should try to describe a bit more detailed in how far your choosen variables "housing condition", "week of age", "T-hobo", "UR%", "vent" and "NH3_ppm" are suitable for the prediction of Ammonia-concentration risk and what are possible other variables that you didn't include? (References)
You could also describe a bit more detailed and underlined with references what health riscs are the most important ones that are correlated to Ammonia-concentration.
Materials and Methods:
How and when were health indicators assessed by whom? Have you done inter observer reliability tests?
Which scale were used to weigh the animals?
What does that mean the remaining birds were housed randomly throughout the room? They run between the cages? Maybe a figure could support the explanation of the setting?
How were the varaibles scaled/coded in which unit?
Discussion
Delete Table 6 there is no relevant information given for the discussion part.
line 2: l instead of t in "Broiler"
line 17: rebuilt sentence...in the production of slow and fast-growing broilers and with low and high stocking density and to correlate risk exposure with health injuries...
lines 21/22: I think stocking density is more commen than housing density? Change in the whole paper
line 163: What does T-hobo and vent mean exactly?
line 173: check if reference has to be stated by number instead of author name (same fore line 179)
line 228: J48 decision tree model for...Table headings and figure subtitles must be labeled in such a way that the content can be understood without the main text. More detailed labeling would be good.
lines 249/250: correlated variables must be explained. What does e.g. bronchi mean? It's not pwas it's feet or more exactly if assessed footpad lesions or ulceration or what ever.
line 268: abbrevations have to be explained by first use IoT (...)
line 286: rebuilt sentence, it doesn't make sence.
line 296: Iin this study a remarkable system was designed, achieving... (rebuilt sentences)
line 300-317: be more precise when comparing results. Use our study resulted in ... like the study from Meier [25]
Or: the results of Hawk et al. [26] are with an accuracy of different to ours ...
It is not always clear at what point your study is meant or a cited one.
line 324: in comparison to [35]? What was their value? But your p-value is'nt significant.
References must be checked. Change et al. by the names of the co-authors, be uniform in the order of year and pages (e.g. 4. vs. 24.)
Author Response
Dear Reviewer, the corrections and answers are in the attac hed file.
Many thanks
Daniella

Reviewer 2 Report
Comments and Suggestions for Authors
The title of the manuscript
is incorrect, 'broided' should be corrected to 'broiler', two abbreviations are used: 1. ammonia concentration in broiler chickens - it would be better in broiler houses; 2. diseases is too much of a generalization.
Citation (left column) should contain the title, not the objective of the study.
There is a lack of the simple summary.
The abstract is too long and does not meet the journal requirements. There are too many details about the methodology. Abbreviations such as SMO should not appear without prior expansion, so the abbreviation should be put in brackets: Sequential Minimal Optimization (SMO).
Introduction
Some statement are not relevant to this research, e.g. text in lines 52-55. However, this too short chapter does not include a references review related to research on the exposure of broiler chickens to ammonia, its various concentrations and related health problems. How do chicken housing systems affect its concentration?
It is important for the authors to distinguish the emission from the concentration of this gas in the poultry house, as they are not the same thing. The work does not focus on emission but on concentration in a given place and time, and this should be emphasized in the introduction.
The journal Animals is intended mainly for scientists involved in animal research, so it would be worth describing in more detail the J48, SMO, etc. models used in the work in the introduction. Why was it worth using them, what do they involve? First of all, how can they contribute to better efficiency of breeding broiler chickens or improve their welfare?
Material and Methods
It is difficult to imagine that the addition of urea in the selected dose allowed the ammonia concentration to be maintained as planned, because this concentration depends on many factors. Have any previous simulations or studies been conducted to develop such a methodology? Can you describe it more precisely?
Line 119, please provide the scale model, device accuracy.
I have concern about the lack of gender division. Maintaining unsexed birds may have allowed sex to determine some parameters, such as body weight and weight gain. Nothing is mentioned about the composition of the feed for different groups of chickens, this has a huge impact on the results, not only the concentration of ammonia. What specific diseases were taken into consideration? At what stage of advancement, and who diagnosed them? It is not only the concentration of ammonia that determines their occurrence.
Results and Discussion
The authors state: ‘Ammonia represents a primary air pollutant in poultry facilities, exerting a significant adverse impact on the ecosystem, the environment, bird welfare, and human health [15]. Therefore, accurately estimating the concentration of NH3 becomes an essential imperative for adequate waste management, aiming to preserve environmental health, human health, and animal welfare [24,15].' This is true, although it should be emphasized that this is only one of very many factors in poultry breeding, the determining of which, and especially mitigation, is necessary to preserve environmental health, human health, and animal welfare.
Conclusions are to general.
References are not prepared according Animals instruction for authors.
Author Response

(The authors gave the same response as above.)

Reviewer 3 Report
Comments and Suggestions for Authors
The manuscript addressed a very appropriate topic for poultry farming; as ammonia in facilities can be a major health problem for birds. However, some points need to be clarified. I actually thought about rejecting the manuscript, without giving it the opportunity to revise; but how do I understand this important topic; I will make more general comments on the main problems; I hope the authors use it to make this manuscript publishable.
1) Summary section. It needs to be re-written, mainly presenting results. In the summary here, over 70% was methodology, which is not suitable for a scientific article. The title talks about correlating with diseases, but in the results of this section nothing appears about this; what diseases appeared? It does not make clear what the levels of ammonia in the environment were like in the different experimental groups. In my opinion, the summary was poor.
2) Introduction section, must clearly report the scientific problem (I see this part present), but the scientific justification for the manuscript is not clear (absent, it is not clear); It was not clear what knowledge gap needed to be researched. Furthermore, the section had short paragraphs (some are not even paragraphs, because they have a single sentence); and the authors should organize this section better.
3) This section is good, with clear information (which is why I recommended the review). In this section I have just one suggestion, which would be related to referencing the groups formed, when growing slowly or quickly; regarding density per m2 (in order to know what is low or high capacity).
4) The captions of tables and figures need to be rewritten and must be self-explanatory.
5) The authors need to present a table with the diet consumed by the animals; as well as the chemical composition. This type of information is essential, even if it is complementary to the article. It is known that the amount of ammonia produced is completely related to the type of diet.
6) The authors must also present data on the animals' real zootechnical performance here; we need to know body weight by phase, by group, as well as food consumption; and calculate feed conversion. Furthermore, I suggest also calculating the food efficiency index. These data are also complementary to this research; but extremely necessary to understand whether the calculations made in different models are reflecting reality. Create a table with this data.
7) The discussion and conclusion section must be revised; after these corrections.
8) Despite my experience in the area, I was unable to gain clarity on the experimental models evaluated; in the data and consequently in its conclusions. I believe that with these changes, I will be able to carry out a much more accurate review.
Author Response

(The authors gave the same response as above.)

Round 2
Reviewer 3 Report
Comments and Suggestions for Authors
The text has improved a lot, but I still have corrections:
a) The authors used commercial food, OK. But the authors need to present a table with the chemical composition of the diet, analyzed.
b) the conclusion section has not had any changes, it remains as in the first version; something very superficial, which in its form does not justify publication, as there is no news. But after reading the work, I understand that the conclusion can be improved considerably.
c) format the manuscript according to the journal’s standards; pay attention to references too.
Author Response
The response is in the file below.
